# Machine Learning in the Management of Lateral Skull Base Tumors: A Systematic Review

**Kotaro Tsutsumi** [1], **Sina Soltanzadeh-Zarandi** [1], **Pooya Khosravi** [1,2], **Khodayar Goshtasbi** [1], **Hamid R. Djalilian** [1,2] **and Mehdi Abouzari** [1,*]

[1] Department of Otolaryngology—Head and Neck Surgery, University of California Irvine, 333 City Blvd. West, Suite 525, Orange, CA 92868, USA

[2] Department of Biomedical Engineering, University of California, Irvine, CA 92617, USA

[*] Correspondence: mabouzar@hs.uci.edu; Tel.: +1-714–509-6096; Fax: +1-714–456–5747

**Abstract:** The application of machine learning (ML) techniques to otolaryngology remains a topic of interest and prevalence in the literature, though no previous articles have summarized the current state of ML application to management and the diagnosis of lateral skull base (LSB) tumors. Subsequently, we present a systematic overview of previous applications of ML techniques to the management of LSB tumors. Independent searches were conducted on PubMed and Web of Science between August 2020 and February 2021 to identify the literature pertaining to the use of ML techniques in LSB tumor surgery written in the English language. All articles were assessed in regard to their application task, ML methodology, and their outcomes. A total of 32 articles were examined. The number of articles involving applications of ML techniques to LSB tumor surgeries has significantly increased since the first article relevant to this field was published in 1994. The most commonly employed ML category was tree-based algorithms. Most articles were included in the category of surgical management (13; 40.6%), followed by those in disease classification (8; 25%). Overall, the application of ML techniques to the management of LSB tumor has evolved rapidly over the past two decades, and the anticipated growth in the future could significantly augment the surgical outcomes and management of LSB tumors.

**Keywords:** lateral skull base surgery; machine learning; artificial intelligence

## 1. Introduction

The application of machine learning (ML) in medicine has significantly grown over the past decade due to the increase in available annotated medical data and computational power [1]. Some of the most prominent studies in the field have included cancer diagnosis and disease predictions [2–5]. ML has also succeeded in capturing meaningful features from images used for classifying pathology slide images, radiographic images, and other clinical imaging modalities for automatic diagnoses [6–10]. Accordingly, the application of ML to medicine will continue to accelerate, with health economists predicting a 10-fold growth in health artificial intelligence (AI) within the next 5 years, saving an annual 150 billion USD for United States healthcare by 2026 [11].

Otolaryngology covers a broad spectrum of subspecialties, presenting unique opportunities to incorporate such ML techniques. For instance, recent advancements have allowed for the automatic detection of abnormalities on tympanic membrane images, recognition of head and neck cancer, and enhancement of cochlear implant performance [10,12–17]. The investigation of various ML applications in otolaryngology remains a topic of interest in the literature [13,14], and this trend is likely to increase in the future.

Previous review articles have summarized the current state of ML application to otolaryngology [13,17,18]. However, the application of ML in the management and diagnosis of lateral skull base (LSB) tumors is yet to be explored. Specifically, these tumors are

related to disorders of hearing, balance, skull base, and facial/vestibulocochlear nerves and constitute a major field within otolaryngology [19,20]. Hence, the application of novel computational techniques to this subspecialty is of great interest. Given such context, this article systematically reviews the published articles regarding ML applications for the diagnosis and management of LSB tumors, along with comments on the future of this field.

## 2. Materials and Methods

Independent searches were conducted on PubMed and Web of Science between August 2020 and February 2021. A unique combination of search terms was devised to identify the relevant literature (Supplemental Table S1). Study inclusion and data extraction are in accordance with the Preferred Reporting Items for Systematic Reviews and Meta-analyses (PRISMA) guidelines (Figure 1) [21]. A total of 53 unique articles were found. Articles deemed nonrelevant on the basis of target pathology and type of technology utilized and those without an available full text were manually eliminated, leaving 32 final articles for inclusion. Articles were categorized into the following categories: MRI tumor segmentation, surgical management, disease classification, and other clinical applications. All articles were evaluated regarding their application task, ML methodology, their outcomes, and potential implications.

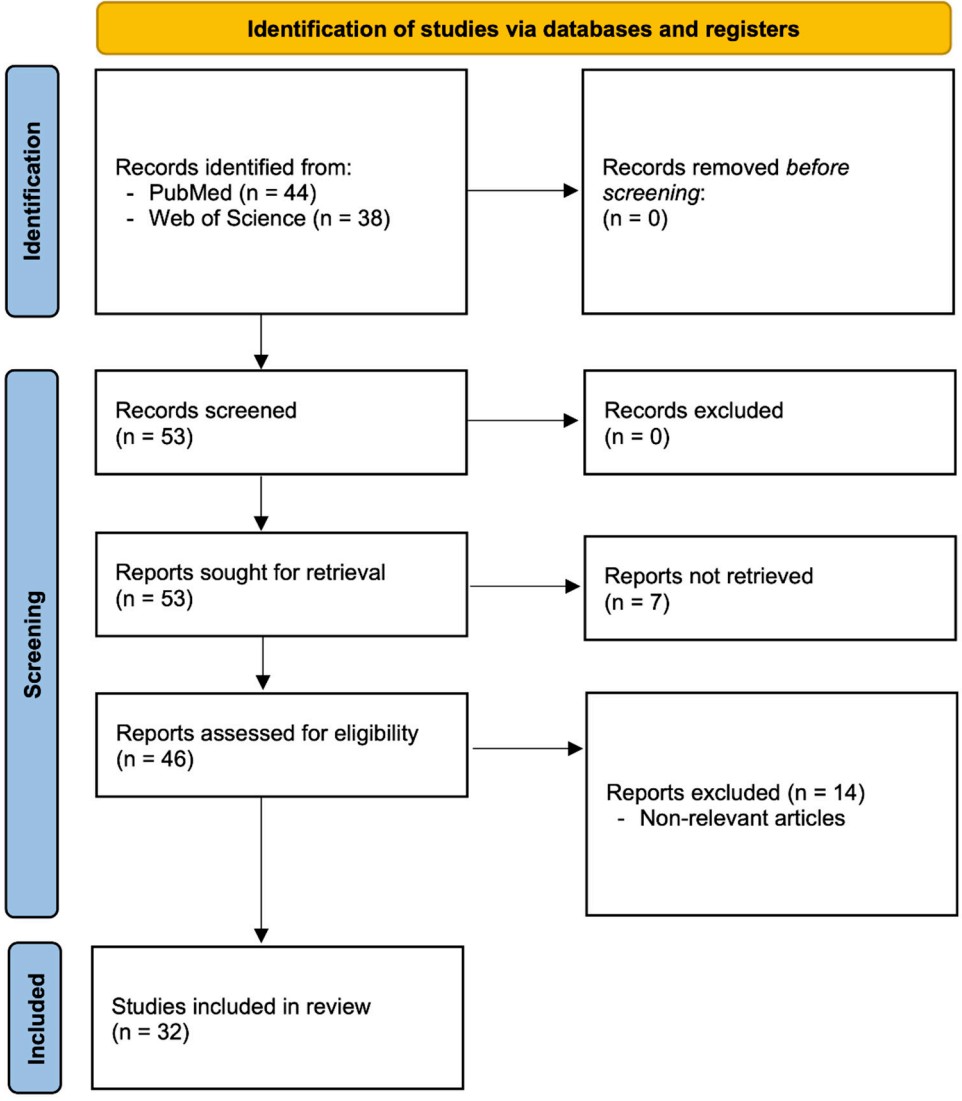

**Figure 1.** PRISMA guidelines outlining search results and excluded articles.

## 3. Results

### 3.1. Topic Trends

The number of articles pertaining to the application of ML to LSB tumor surgeries has significantly increased since the first article in the field was published in 1994 (Figure 2). Most articles were included in the category of surgical management (13; 40.6%), followed by those in disease classification (8; 25%), MRI tumor segmentation (8; 25%), and others (3; 9.4%). The most commonly employed ML category was tree-based algorithms (Figure 3).

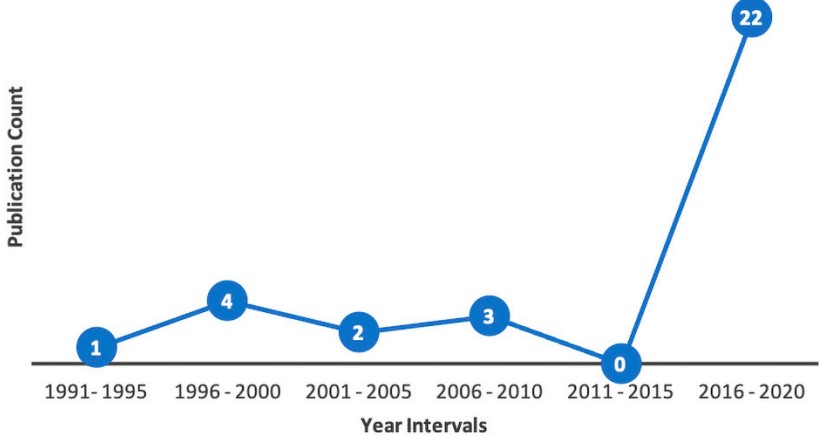

**Figure 2.** Publication count per time interval of all articles included in the review.

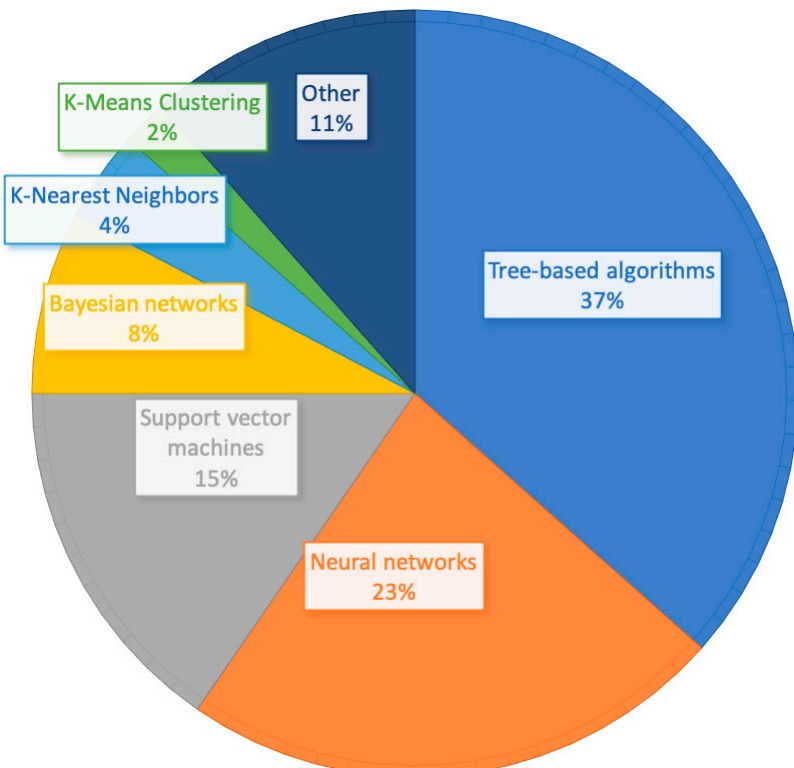

**Figure 3.** Frequency of ML techniques employed among the articles included in the review.

### 3.2. Surgical Management

LSB tumor surgeries can be associated with major complications and should be pursued after careful consideration of alternative management strategies [22]. In one of the earliest uses of ML regarding the management of LSB tumors, Telian identified the probability of vestibular schwannoma (VS) enlargement via a decision tree algorithm [23].

Furthermore, when pursuing surgery, identifying at-risk patients and optimizing the pre-, peri-, and post-operative parameters is important. Hence, many studies have applied ML for predicting surgical management-related factors, including post-surgery complications and mortality (Table 1).

**Table 1.** Summary of manuscripts with machine learning applications for surgical management.

| Author, Year, References | Aim | Algorithm(s) | Outcomes |
|---|---|---|---|
| Abouzari et al., 2020 [24] | Predicting the recurrence of vestibular schwannoma using artificial neural network compared to logistic regression. | Artificial neural network | Artificial neural network was superior to logistic regression in predicting the recurrence of vestibular schwannoma with an accuracy of 0.70. |
| Claudia et al., 2019 [25] | Using machine-learning radiomics to predict response to CyberKnife treatment of vestibular schwannoma. | Decision tree Random forest XGBoost | Machine-learning radiomics predicted response to CyberKnife treatment of vestibular schwannoma with an accuracy of 0.92. |
| D'Amico et al., 2018 [26] | Computing quantitative biomarkers from MRI to predict CyberKnife treatment response on vestibular schwannoma. | Decision tree | Treatment response was predicted with an accuracy of 0.85 using a machine-learning based radiomic pipeline. |
| Cha et al., 2020 [27] | Predicting hearing preservation following surgery in patients with vestibular schwannoma. | Support vector machine Gradient boosting machine Deep neural network Random forest | Hearing preservation was predicted with most accurately by deep neural network with an accuracy of 0.9 |
| Dang et al., 2021 [28] | Elucidation of risk factors contributing to increased length of stay after vestibular schwannoma surgery. | Random forest | Preoperative tumor volume and dimensions, coronary artery disease, hypertension, major complications, and operative time were significant predictive factors for prolonged length of stay |
| George-Jones et al., 2020 [29] | Predicting post-stereotactic surgery enlargement of vestibular schwannoma. | Support vector machine | Enlargement was predicted with an overall AUC of 0.75. |
| Langenhuizen et al., 2020 [30] | Prediction of tumor progression after stereotactic surgery of vestibular schwannoma using MRI texture. | Support vector machine | Machine learning achieved an AUC of 0.93 and an accuracy of 0.77 for prediction of tumor progression. |
| Langenhuizen et al., 2020 [31] | Prediction of transient tumor enlargement after vestibular schwannoma radiosurgery using MRI textures. | Support vector machine | A maximum AUC of 0.95, sensitivity of 0.82, and specificity of 0.89 were achieved for prediction. |
| Langenhuizen et al., 2019 [32] | Predicting the influence of dose distribution on the treatment response of gamma knife radiosurgery on vestibular schwannoma. | Support vector machine | 3D histogram of oriented gradients features correlate with treatment outcomes (AUC = 0.79, TPR = 0.80, TNR = 0.75, with support vector machine) |
| Langenhuizen et al., 2018 [33] | Using MRI texture feature analysis to predict vestibular schwannoma gamma knife radiosurgery treatment outcomes. | Support vector machine Decision tree | Treatment outcomes were predicted with an accuracy of 0.85 using machine learning. |
| Lee et al., 2016 [34] | Predicting risk factors leading to communicating hydrocephalus following gamma knife radiosurgery for vestibular schwannoma. | K-nearest neighbors classifier Support vector machine Decision tree Random forest AdaBoost Naïve bayes Linear discriminant analysis Gradient boosting machine | Age, tumor volume, and tumor origin are significant predictors of communicating hydrocephalus. Developing communicating HCP following gamma knife radiosurgery is most likely if the tumor was of vestibular origin and had a volume $\geq 13.65$ cm$^3$. |
| Telian et al., 1994 [23] | Management of vestibular schwannoma between 5–15 mm | Decision tree | Most important factor in determining to proceed with surgery is the probability of tumor growth. |
| Yang et al., 2020 [35] | Prediction of progression/outcome of vestibular schwannoma after gamma knife radiosurgery using MRI data | Support vector machine | Machine learning predicted long-term outcome and transient pseudoprogression with an accuracy of 0.88 and 0.85, respectively. |

One major complication of VS surgery is hearing loss, and in order to improve patient outcomes, ML has recently been explored as a method to predict the rate of hearing preservation after VS surgery. Notably, Cha et al. compared support vector machine (SVM), gradient boosting machine (GBM), deep neural network (DNN), and diffuse random forest algorithms to predict hearing preservation following VS surgery [27]. DNN showed the greatest promise, achieving an accuracy of 90% and a sensitivity and specificity of 93% and 86%, respectively. ML has been utilized to identify the major risk factors associated with developing communicating hydrocephalus (HCP) following gamma knife radiosurgery (GKRS) of VS. In a cohort of 702 patients, Lee et al. identified that a vestibular nerve origin

tumor with a volume $\geq 13.65$ cm$^3$ is a major risk factor of developing communicating HCP following GKRS of VS using several ML classifiers, most of which performed with an accuracy above 94% for the classification tasks [34]. In order to predict the recurrence of VS, Abouzari et al. compared the use of an artificial neural network (ANN) and logistic regression on 789 patients [24]. ANN showed a superior performance with an accuracy of 70%, sensitivity of 61%, and specificity of 81% compared to an accuracy of 56%, sensitivity of 44%, and specificity of 69% for the logistic regression statistical model. Dang et al. elucidated risk factors for predicting an increased length of stay after the resection of VS [28]. Using a random forest model, they found that the preoperative tumor volume and dimensions, coronary artery disease, hypertension, major complications, and operative time were significant predictive factors for prolonged length of stay after surgery.

Recent studies have shown the efficacy of inputting magnetic resonance imaging (MRI) characteristics into ML algorithms in predicting surgical outcomes. Langenhuizen et al. investigated the application of ML in GKRS of VS in three studies [30,32,33]. Particularly, they investigated cases with significant tumor regression within one year following GKRS compared to cases with continued tumor progression following the intervention. Utilizing SVM, Langenhuizen et al. classified these two classes with an accuracy of 85% and a maximum area under the curve (AUC) of 0.95. Langenhuizen et al. also used MRI texture features in predicting tumor progression after stereotactic radiosurgery [31]. MRI texture information was used to train an SVM, achieving an AUC of 0.93 and accuracy of 77%. Similarly, George-Jones et al. input MRI texture and shape features from 53 patients into a SVM to predict the post-stereotactic surgery enlargement of VS, achieving an overall AUC of 0.75 [29].

Yang et al. utilized radiomic analysis on pre-radiosurgical MRI data to predict the pseudoprogression and long-term outcome of VS after GKRS [35]. The raw dataset consisted of pre-GKRS MRI data for 336 patients and a median follow-up period of 65 months. After radiomic features were generated in a two-level binary classification model, a SVM was trained based on the selected features to classify the data into three response groups (tumor non-response, tumor regression with no pseudoprogression, and tumor regression with pseudoprogression). The trained model predicted the long-term outcome with an accuracy of 88.4% and transient pseudoprogression with an accuracy of 85.0%. D'Amico et al. explored the use of ML to predict the response to CyberKnife treatment of VS in 38 patients using various approaches [25,26]. Notably, biomarker quantifications extracted from patient MR images were used as the input to the algorithms. Among the algorithms explored, the decision tree showed the greatest promise, achieving a 92% accuracy.

### 3.3. Disease Classification

The emerging shortage of otolaryngology expertise worldwide can significantly impact the appropriate diagnosis and management of LSB tumors [36]. Accordingly, ML has been applied to classify various neurotologic diseases based on patient presentations (Table 2). Nouraei et al. developed a Gaussian Process Ordinal Regression Classifier, a type of Bayesian classifier, for identifying true vs. suspicious VS cases [37]. Clinical and audiometric information were input into the model, which ultimately achieved an AUC of 0.80.

A research team based at the University of Tampere and Helsinki University Hospital published seven articles aimed to classify VS, benign positional vertigo, Meniere's disease, sudden deafness, traumatic vertigo, and vestibular neuritis [38–44]. Miettinen et al. and Juhola et al. utilized the Bayesian probabilistic model, k-nearest neighbors classifier, discriminant analysis, Naïve Bayesian decision rule, k-means clustering, decision trees, neural networks, and Kohonen networks [38,39,43]. The inputs consisted of 38 otology and neurotology attributes, and the Bayesian probabilistic model attained the best performance with an accuracy of 97%. Laurikkala et al. and Kentala et al. performed this classification task using genetic algorithm-based ML systems, attaining an accuracy of above 90% and 80% for detecting the correct pathologies, respectively [41,42]. Kentala et al. and Viikki et al.

used decision trees to classify the aforementioned six disease processes, and the accuracy for detecting each pathology was above 90% for both studies [40,44].

**Table 2.** Summary of the manuscripts with machine learning applications for disease classification.

| Author, Year, References | Aim | Algorithm(s) | Outcomes |
|---|---|---|---|
| Juhola et al., 2008 [38] | Classification of otoneurological diseases including vestibular schwannoma, benign positional vertigo, Ménière's disease, sudden deafness, traumatic vertigo, and vestibular neuritis given patient attributes. | K-nearest neighbors classifier<br>Discriminant analysis<br>Naïve bayes<br>K-means clustering<br>Decision trees<br>Neural networks<br>Kohonen networks | Discriminant analysis performed the best with an average accuracy of 0.96. |
| Juhola et al., 2001 [39] | Classification of otoneurological diseases including vestibular schwannoma, benign positional vertigo, Ménière's disease, sudden deafness, traumatic vertigo, and vestibular neuritis given patient attributes. | Kohonen networks | The model attained a maximum accuracy of 0.98 for classification of overrepresented pathologies. |
| Kentala et al., 2000 [40] | Classification of otoneurological diseases including vestibular schwannoma, benign positional vertigo, Ménière's disease, sudden deafness, traumatic vertigo, and vestibular neuritis given patient attributes. | Decision tree | The decision tree achieved an accuracy between 0.94 and 1 according to different pathologies. |
| Kentala et al., 1999 [41] | Classification of otoneurological diseases including vestibular schwannoma, benign positional vertigo, Ménière's disease, sudden deafness, traumatic vertigo, and vestibular neuritis given patient attributes. | Genetic algorithm | The genetic algorithm attained an accuracy of 0.80. |
| Laurikkala et al., 2001 [42] | Classification of otoneurological diseases including vestibular schwannoma, benign positional vertigo, Ménière's disease, sudden deafness, traumatic vertigo, and vestibular neuritis given patient attributes. | Genetic algorithm | The machine learning model attained an accuracy of 0.90. |
| Miettinen et al., 2008 [43] | Classification of otoneurological diseases including vestibular schwannoma, benign positional vertigo, Ménière's disease, sudden deafness, traumatic vertigo, and vestibular neuritis given patient attributes. | Bayesian classifier | The Bayesian classifier attained an accuracy of 0.97. |
| Viikki et al., 1999 [44] | Classification of otoneurological diseases including vestibular schwannoma, benign positional vertigo, Ménière's disease, sudden deafness, traumatic vertigo, and vestibular neuritis given patient attributes. | Decision tree | An average accuracy of over 0.95 was achieved. |
| Nouraei et al., 2007 [37] | Identification of vestibular schwannoma cases from a population suspected to harbor the tumor. | Bayesian classifier | The machine learning algorithm achieved an AUC of 0.80 for classification. |

### 3.4. Tumor Segmentation

Radiographic images constitute a key component of LSB tumor management. MRIs in particular are useful for a diagnosis, surgical planning, and follow-up of such tumors [45]. Consequently, many image-based ML studies have focused on detecting tumors on MRI scans (Table 3), with Dickson et al. publishing the first paper in 1997 [46]. An ANN was trained and tested on MRI scans from 50 patients for the detection of VS, achieving an overall false-negative rate of 0% and false positive rate of 8.6%. Lee et al. developed a two-pathway U-Net model for the automatic delineation of VS from multiparametric MRI scans [47]. The model was trained on T1-weighted (T1W), T2-weighted (T2W), and T1W with contrast (T1W/C) MRI scans from 516 patients, achieving an average dice score of 0.90 ± 0.05. Similarly, George-Jones et al. trained a U-Net on 130 T1W/C MRI scans from 65 patients for the segmentation of VS [48]. The segmentation algorithm was compared to manual segmentation constructed by a clinician, yielding an interclass correlation coefficient of 0.99. The algorithm was further used to detect the growth of tumors after at least 5 months since the initial scan. CNN performed superior to the greatest linear dimension for growth detection in terms of sensitivity, specificity, and AUC. Furthermore, Shapey et al. developed a CNN for segmentation of VS on T2W and T1W/C MRI scans [49]. The algorithm contained a computational attention module that enables the

CNN to focus more on the target region of interest. The dice score for segmentation was 0.93 and 0.94 for T1W/C and T2W, respectively. Lee at al. used a U-Net-based dual-pathway model for the segmentation of VS from MRI scans of 861 patients who underwent GKRS, which performed with an average dice score of 0.90 [50]. They further utilized this model to demonstrate its viability in measuring tumor growth or regression longitudinally.

**Table 3.** Summary of the manuscripts with machine learning applications for MRI/CT tumor segmentation.

| Author, Year, References | Aim | Algorithm(s) | Outcomes |
|---|---|---|---|
| Dickson et al., 1997 [46] | Detection of vestibular schwannoma on MRI scans. | Artificial neural network | The neural network attained a false negative rate of 0 and false positive rate of 0.086. |
| George-Jones et al., 2020 [29] | Segmentation of vestibular schwannoma from T1W with contrast MRI scans. | U-Net | The model achieved an interclass correlation coefficient of 0.99. |
| Lee et al., 2021 [50] | Segmentation of vestibular schwannoma from MRI scans. | U-Net | The model performed with an average dice score of 0.90. |
| Lee et al., 2020 [47] | Segmentation of vestibular schwannoma from multiparametric MRI scans. | U-Net | The U-Net delineated vestibular schwannoma with a dice score of 0.90 ± 0.05. |
| Neves et al., 2021 [51] | Segmentation of temporal bone structures from CT scans. | AH-Net U-Net ResNet | The model's performed with dice scores of 0.91, 0.85, 0.75, and 0.86 for inner ear, ossicles, facial nerve, and the sigmoid sinus, respectively. |
| Shapey et al., 2021 [49] | Segmentation of vestibular schwannoma from T2W and T1W with contrast MRI scans. | Convolutional neural network | By employing a computational attention module, the algorithm attained a dice score of 0.93 and 0.94 for T1W and T2W, respectively. |
| Uetani et al., 2020 [52] | Denoising of MRI for high spatial resolution-MR cisternography for cerebellopontine angle legions via deep learning-based reconstruction. | Convolutional neural network | Images reconstructed with deep learning-based reconstruction had higher image quality ($p < 0.001$) due to reduced image noise while maintaining contrast and sharpness. |
| Windisch et al., 2020 [53] | Segmentation of vestibular schwannoma or glioblastoma from T1W, T2W, and T1W with contrast MRI scans with a focus on the explainability. | Convolutional neural network | The model achieved an accuracy of 0.93 while the Grad-CAM software also showed it correctly focused on tumor loci. |

A commonly cited issue regarding automation of clinical technology is its low transparency. Aiming to alleviate this issue, Windisch et al. focused on the explainability of ML algorithms [53]. The study trained a CNN on T1W, T2W, and T1W/C MRI scans from 1223 patients who either had VS or glioblastoma. Grad-CAM, a software that highlights segments of an image deemed important by a classification algorithm, was used simultaneously. The model achieved an accuracy of 93%, while Grad-Cam showed that the model was focusing correctly on the tumor for making predictions.

Uetani et al. evaluated the efficacy of deep learning-based reconstruction (DLR) for denoising MRI scans for high spatial resolution (HR)-MR cisternography for cerebellopontine angle lesions [52]. Compared to ordinary frequency-based models, DLR with a separate path for high-frequency components that are processed as feature maps in the feature conversion layers is able to learn the CNN parameters to reduce noise without sacrificing spatial resolution. In the retrospective study of 35 patients who underwent HR-MR cisternography, images reconstructed with and without DLR were compared based on the signal-to-noise ratio (SNR) and contrast of cerebrospinal fluid and pons; the sharpness of the normal-side trigeminal nerve using full width at half maximum (FWHM); and a qualitative score of noise quality, sharpness, artifacts, and overall image quality. They found that the image with DLR had a higher image quality ($p < 0.001$) while maintaining contrast and sharpness.

In addition to MRIs, computed tomography (CT) scans have been targeted as a mode for tumor segmentation tasks. Neves et al. developed a ML algorithm based on three CNN models, including AH-Net, U-Net, and ResNet, for the segmentation of temporal bone structures [51]. The model's performance was comparable to manual segmentation by otology specialists, with a dice score of 0.91, 0.85, 0.75, and 0.86 for the inner ear, ossicles, facial nerve, and the sigmoid sinus, respectively.

### 3.5. Other Clinical Applications

Many studies have applied ML to other unique aspects of LSB tumor management (Table 4). Surgical pose estimation is the task of extracting the pose, which is defined by the coordinates, forward angle, projection angle, and depth of the instrument, from an X-ray image [54]. Kugler et al. created i3PosNet, a CNN-powered instrument pose estimation method capable of high-precision estimations of poses, specifically for applications in temporal bone surgery. Three datasets were used to train and evaluate i3PosNet (synthetic radiographs with medical screws, synthetic radiographs with other instruments, and real X-ray images of screws) to show that i3PosNet generalizes real X-ray images while only trained on synthetic data and that it generalizes other instruments. The trained i3PosNet was found to estimate the pose with an error <0.05 mm on synthetic data, outperforming other state-of-the-art registration methods. Regardless, the application to real data remained a challenge due to the unavailability of large representative datasets.

**Table 4.** Summary of the manuscripts with machine learning applications in other topics.

| Author, Year, References | Aim | Algorithm(s) | Outcomes |
|---|---|---|---|
| Chang et al., 2019 [55] | Prediction of cochlear dead region prevalence given various sensorineural hearing loss patient data. | Decision tree Random forest | The random forest and the classification tree were capable of predicting cochlear dead regions by an accuracy of 0.82 and 0.93, respectively, while illustrating strong predictive factors for cochlear dead region prevalence. |
| Rasmussen et al., 2018 [56] | Elucidation of perilymph proteins associated to vestibular schwannoma related hearing loss and tumor diameter. | Random forest | A perilymph protein, alpha-2-HS-glycoprotein (P02765) was determined to be an independent variable for predicting tumor-associated hearing loss. |
| Kügler et al., 2020 [54] | Creation of a convolutional neural network-powered pose estimation method capable of high-precision estimation of poses for application to temporal bone surgery. | Convolutional neural network | The instrument was found to estimate the pose, or an estimation of location of surgical instrument using an X-ray image, with an error <0.05 mm on synthetic data. |

Chang et al. used a classification tree and random forest to predict the prevalence of cochlear dead regions given various sensorineural hearing loss (SNHL) patient data [55]. The random forest and the classification tree predicted cochlear dead regions with an accuracy of 82% and 93%, respectively, and found that word recognition scores, hearing thresholds at each frequency, etiology of VS or Meniere's disease, and frequency parameters are strong factors for predicting prevalence of cochlear dead regions.

Rasmussen et al. used ML techniques to elucidate the association between perilymph proteins and VS related hearing loss and tumor diameter [56]. A univariate linear regression and random forest were used to predict the hearing loss and tumor diameter given the perilymph proteome levels. Variable importance of each protein was used to illustrate which perilymph protein is significantly correlated to tumor-associated hearing loss, and the results showed that Alpha-2-HS-glycoprotein (P02765) was an independent predictor of tumor-associated hearing loss.

### 4. Discussion

This systematic review aims to comprehensively investigate studies that apply ML techniques in managing LSB tumors. Undiagnosed tumors, mismanaged disease process, and post-treatment adverse outcomes undoubtedly place a mental and physical toll on patient [57]. It has also been estimated that rehospitalization post-discharge costs 17 billion USD annually in avoidable Medicare expenditures [58]. Patients with head and neck cancer have been identified as a high-risk group for readmission, with rates ranging between 6% and 26.5% [59–62]. Many studies have pointed out the likeliness of the Hospital Readmissions Reduction Program (HRRP) program expanding with the aim to reduce such avoidable expenditures [63,64]. However, the HRRP has opened doors to controversies regarding readmission rates and reimbursements among other fields to which it has been implemented [65]. Developing high-quality predictive models could mitigate these issues

by enabling physicians to devise a more personalized treatment plan. Consequently, the results from many of the surgical management studies pertaining to LSB tumors could be utilized for developing a stratified model in which hospitals could be penalized for cases that were predicted to have low probability post-surgical incidences.

Similar predictive indices have been considered before, such as the LACE+ index or the University of Kansas Health Systems and Atrium Health predictive programs [66,67]. These programs analyze the clinical and socioeconomic factors for predicting patients' probability of readmission and have successfully reduced their readmission rates; however, neither employed ML techniques. Incorporating the predictive capacity of ML algorithms into such frameworks could improve the quality of these models by potentially increasing their predictive capacity while also illuminating clinical variables that may be important risk factors for readmission.

ML has also led to advancements in basic science, as seen from the study by Rasmussen et al. [56]. They presented that the Alpha-2-HS-glycoprotein is highly correlated to the onset of VS-related hearing loss by examining the process by which ML classifies VS patients with SNHL from those without it. This is fundamentally different from traditional mass spectrometry studies of the perilymph in that it uncovered an independent predictor of VS-related hearing loss that is not readily apparent by the quantification and composition of the perilymph. As such, ML offers a new dimension of utility by which algorithms classify cases based on their own criteria, uncovering novel independent variables for the classified cases. This use of ML is only possible because it conducts classification according to its own unique criteria that is dissimilar to the approach of human researchers. Given such an approach, combined with the advent of novel algorithms such as AlphaFold for protein folding predictions, ML is bound to revolutionize how researchers approach basic science [68].

The ML technology elaborated on within this current study included those that are prominently utilized among clinical informatics, notably including tree-based algorithms, neural networks, and SVMs. Tree-based algorithms were most widely utilized among studies included in the current review, most likely given the relative simplicity of the algorithm mechanism, along with the capacity for it to easily extract feature importance when predicting an outcome. ML applications in healthcare require a high-level of transparency, and these advantages of tree-based algorithms may have been a deciding factor in choosing which algorithms to employ. Furthermore, advancements in neural network technology have been revolutionary in conducting image classification and segmentation tasks. Though primitive versions of such technology have existed for many years, their utility has truly been proven since the successful implementation of graphics processing units around 2010 that enabled accelerated training, explaining the rapid increase in studies published between 2016 and 2020, mostly consisting of those utilizing CNNs [69].

However, the technology in this field has been constantly growing, and additional innovative algorithms have emerged over the past several years. For instance, generational adversarial networks have been employed for unsupervised image classification tasks, while transformers have been proven to be useful for natural language processing tasks [70,71]. While many of the algorithms included in the current study and those not included as aforementioned are all very effective algorithms, the user must carefully select the most effective algorithm according to their project's goals. For instance, CNN has been successfully applied for image classification tasks, though it may not function well as a generic classifier for tabular data when compared, for instance, to SVMs. Subsequently, more discourse regarding different ML algorithms and how they can be applied to different tasks in healthcare must be held.

While the application of ML shows great promise, its use in clinical management has several limitations, primarily regarding data accuracy. For instance, many databases that outline patient characteristics are often subject to errors and could have missing values or variables [72]. For these to be input into an ML algorithm, the missing data must

be estimated via various software, and results obtained through such processes must be interpreted with caution.

Furthermore, Windisch et al. expressed that the explainability of ML algorithms is critical for the actual translation of such technology into clinics [53]. Techniques such as Grad-CAM, as used in this study, exist to highlight particular areas of an image considered important through an image classification process but does not provide a comprehensive explanation of how each aspect of the image precisely influences the prediction [73]. ML explainability and interpretation is a topic that is under extensive investigation. McGrath et al. pointed this out in their study, claiming the limited clinical use of the complete automation of tumor segmentation on MRI scans due to insufficient validation in a clinical setting and, in turn, proposed a semi-automated method that leaves a margin for physician control that still performed significantly better than the manual approach [74]. Consequently, as much as it is important to develop a novel technology that pushes the forefront of medical informatics, it is also important to consider how such technology can be applied in clinical practice.

## 5. Conclusions

This study provides a comprehensive review of studies that apply ML techniques to various aspects of the diagnosis or management of lateral skull base disease processes. Emerging studies utilized ML algorithms to investigate surgical treatment, disease classification, tumor segmentation, and other clinical applications pertaining to lateral skull base disease processes. As the number of ML studies in the field of skull base surgery increases, this study provides a valuable overview of the current literature for the readership.

**Supplementary Materials:** The following supporting information can be downloaded at https://www.mdpi.com/article/10.3390/ohbm3040007/s1: Table S1: Detailed search strategies used to retrieve articles.

**Author Contributions:** Conceptualization: K.T., S.S.-Z., K.G., H.R.D. and M.A.; methodology: K.T., S.S.-Z., P.K., K.G. and M.A.; formal analysis: K.T, S.S.-Z. and P.K. investigation: K.T, S.S.-Z., P.K. and K.G.; resources: K.T. and S.S.-Z.; writing—original draft preparation: K.T., S.S.-Z., K.G. and M.A.; writing—review and editing: K.T., S.S.-Z., K.G., H.R.D. and M.A.; supervision: H.R.D. and M.A.; and project administration: K.T. and M.A. All authors have read and agreed to the published version of the manuscript.

**Funding:** This research received no external funding.

**Institutional Review Board Statement:** Not applicable.

**Informed Consent Statement:** Not applicable.

**Conflicts of Interest:** The authors declare no conflict of interest. Mehdi Abouzari was supported by the National Center for Research Resources and the National Center for Advancing Translational Sciences, National Institutes of Health, through Grant TL1TR001415.

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
