# Peer review of "Machine Learning in the Management of Lateral Skull Base Tumors: A Systematic Review"

_2504-463X, doi:10.3390/ohbm3040007_

Round 1

Reviewer 1 Report

The article is a review paper. In methods based on machine learning, an important element is the learning and validation set. The effectiveness of the prediction and the thesis depends largely on the selection of criteria and the amount of data and the appropriate selection of data. 

Minor remarks:

1) I suggest describing in more detail on what basis and from what the scope of selection of input data for the research problem posed in terms of the results achieved by the method. 

2) The authors could refer more extensively to other methods not included in the comparative analysis.

3) The article needs minor language corrections.

Author Response

We greatly appreciate your time in reviewing our manuscript and for the opportunity to respond to your comments.

Response 1) We appreciate this important comment. In order to clarify this point, the following has been incorporated into the “Materials and Methods” section:
Lines 66-74: “Articles deemed non-relevant on the basis of target pathology and type of technology utilized and those without an available full text were manually eliminated, leaving 32 final articles for inclusion.”

Response 2) Thank you for this important comment. We included the following commentary in the “Discussion” section in order to elaborate on this point:
Lines 304-316: “However, technology in this field has been constantly growing, and additional innovative algorithms have emerged over the past several years. For instance, generational adversarial networks have been employed for unsupervised image classification tasks, while transformers have been proven to be useful for natural language processing tasks.70,71 While many of the algorithms included in the current study and those not included as aforementioned are all very effective algorithms, the user must carefully select the most effective algorithm according to their projects’ goals. For instance, CNN has been successfully applied for image classification tasks, though it may not function well as a generic classifier for tabular data when compared, for instance to SVMs. Subsequently, more discourse regarding different ML algorithms and how they can be applied to different tasks among healthcare must be held.”

Response 3) We appreciate this comment and have made edits to the manuscript where relevant.

Reviewer 2 Report

1.In the Results section, the interpretation of Figure 2 needs to be clear. In the figure 2, since 2011, there has been a huge increase in the number of articles; the number of articles has remained low for 20 years before that. The reason of this change can be explained in the discussion.

 2.This figure 2 has 29 articles, does it mean that the application of ML in LSB tumor surgery is 29?

3.In the Results section, the authors say that: “The most commonly employed ML category was tree-based algorithms.” Please give reasons for this phenomenon.

 4. In the Discussion section, please provide more details about the sentence: “Incorporating the predictive capacity of ML algorithms into such frameworks could improve the quality of these models”.

 5. In this manuscript, the authors provide a comprehensive review of studies that apply ML techniques to various aspects of lateral skull base disease processes. It is recommended that the authors add some algorithm recommendations in the discussion section, which may be more helpful to readers.

Author Response

We greatly appreciate your time in reviewing our manuscript and for the opportunity to respond to your comments.

Response 1) We appreciate this important comment. In order to clarify this point, the following has been incorporated into the “Discussion” section:
Lines 298-303: “Furthermore, advancements in neural network technology has been revolutionary in conducting image classification and segmentation tasks. Though primitive versions of such technology have existed for many years, its utility has truly been proven since the successful implementation of graphics processing units around 2010 that enables accelerated training, explaining the rapid increase in studies published between 2016 and 2020, mostly consisting of those utilizing CNNs.69

Response 2) Thank you for this important comment. The corrections have been made to Figure 2 and has been incorporated to the manuscript.

Response 3) Thank you for this thoughtful comment. The following has been added into the “Discussion” section in order to address this topic:
Lines 292-299: “ML technology elaborated within this current study include those that are prominently utilized among clinical informatics, notably including tree-based algorithms, neural networks, and SVMs. Tree-based algorithms were most widely utilized among studies included in the current review, most likely given the relative simplicity of the algorithm mechanism along with the capacity for it to easily extract feature importance when predicting an outcome. ML application to healthcare requires a high-level of transparency, and these advantages of tree-based algorithms may have been a deciding factor in choosing which algorithms to employ.”

Response 4) Thank you for this comment. The following has been added into the “Discussion” section in order to address this issue:
Lines 275-278: “Incorporating the predictive capacity of ML algorithms into such frameworks could improve the quality of these models by potentially increasing their predictive capacity while also illuminating clinical variables that may be important risk factors for readmission.”

Response 5) Thank you for this important comment. The following has been added to the “Discussion” section in order to elaborate on this topic:
Lines 305-317: “However, technology in this field has been constantly growing, and additional innovative algorithms have emerged over the past several years. For instance, generational adversarial networks have been employed for unsupervised image classification tasks, while transformers have been proven to be useful for natural language processing tasks.70,71 While many of the algorithms included in the current study and those not included as aforementioned are all very effective algorithms, the user must carefully select the most effective algorithm according to their projects’ goals. For instance, CNN has been successfully applied for image classification tasks, though it may not function well as a generic classifier for tabular data when compared, for instance to SVMs. Subsequently, more discourse regarding different ML algorithms and how they can be applied to different tasks among healthcare must be held.”